# Atmospheric Precipitable Water Vapor and its Correlation with Clear Sky Infrared Temperature Observations

Vicki Kelsey[1], Spencer Riley[2], and Kenneth Minschwaner[2]

[1]Langmuir Laboratory for Atmospheric Research, New Mexico Institute of Mining and Technology, Socorro, NM 87801 USA
Now at the Atmospheric and Environmental Sciences Program, South Dakota School of Mines and Technology, Rapid City, SD 57701 USA
[2]Department of Physics, New Mexico Institute of Mining and Technology, Socorro, NM 87801 USA

**Correspondence:** Vicki Kelsey (vicki.kelsey@mines.sdsmt.edu)

**Abstract.** Precipitable water vapor (PWV) is the vertically integrated amount of water vapor in the atmosphere, and it is a valuable predictor for weather forecasting. Currently, the use of sophisticated instrumentation can limit the number of PWV measurement sites, which affects the accuracy of forecast models in regards to storm formation, strength, and the potential for precipitation. We have analyzed relationships between PWV and zenith sky temperature measurements for the dry climate zone found in the North American Desert Southwest, specifically over Socorro, New Mexico (34°N, 107°W). Daily measurements of the ground and zenith sky temperatures have been made at Socorro for two complete annual cycles using low-cost infrared thermal sensors. Radiosonde measurements of PWV from National Weather Service stations located in nearby Albuquerque, and Santa Teresa, New Mexico, are input into our dataset and analysed via a newly developed computational tool. Our results show that an exponential relationship between PWV and zenith sky temperature holds for the Desert Southwest, but with parameters that are different than those obtained previously over the more moist climate zone of the North American Gulf Coast. Model simulations can accurately reproduce the observed relationship between PWV and temperature, and the results suggest that half of the signal in temperature is directly related to changes in opacity due to changes in PWV, while the other half is due to changes in air temperature that usually accompany changes in PWV.

## 1   Introduction

The amount of water in the atmosphere is an important factor that can, along with other factors, determine the amount of rainfall and influence the dynamical evolution of convective storms. Weather forecasting is dependent upon having accurate precipitable water vapor (PWV) data with sufficient temporal/spatial coverage over the forecasting area (Yang and Smith, 2018; Marcus et al., 2007). Increasing the availability of PWV data will ensure more accurate forecasts; especially in higher elevation arid climate zones where there are large distances between existing PWV measurement sites (Maussion et al., 2014; Chen et al., 2018; Zhao et al., 2019). Although PWV can be obtained from infrared measurements on satellite platforms such as GOES-R (Schmit et al., 2018, 2017), potentially large observation angles can result in degraded spatial resolution, and may not provide adequate information for numerical weather prediction models to take into account local variations in PWV.

PWV strongly influences atmospheric dynamics. This is most evident in the fact that when large amounts of PWV are

observed, there is a greater probability for uplifting convection and cloud formation (Raj et al., 2004). This leads to applications
in numerical weather prediction (Wang et al., 2007), as well as climate change modeling and analysis (Gradinarsky et al., 2002).
When the vertically integrated amount of water vapor is more than twice the climatological amount, heavy precipitation can
occur (Wang et al., 2017), which may lead to soil erosion and flooding. Higher amounts of PWV tend to be located near the
equator and especially near Intertropical Convergence Zones, with a general decrease in PWV from low to high latitudes (Raj
et al., 2004).

In this paper we use the standard definition of PWV (Salby, 1996), which is determined by the integrated amount of water
vapor that is contained in a vertical column of air extending from the Earth's surface to the top of the atmosphere, typically
expressed as the height of the liquid water equivalent. In clear skies (the focus of this work), all of this water is in vapor form
and the expression becomes

$$\text{PWV} = \frac{1}{\rho g} \int\limits_0^{p_o} \mu(p) dp, \tag{1}$$

where $\rho$ is the mass density of liquid water, $g$ is the acceleration of gravity, $\mu(p)$ is the mass mixing ratio of water vapor, and
the integral is over pressure $p$ from zero to some surface pressure $p_o$.

Typically, the water vapor mass density decreases quasi-exponentially with increasing altitude (decreasing pressure), such
that the majority of the total column is near the surface. Previous studies have determined that 40% to 60% of the contribution
to sea-level PWV occurs in the pressure layer between 1000 3 hPa and 850 hPa, with roughly 90% lying between the surface
and 500 hPa (Ross and Elliott, 1996; Wang et al., 2007; Raj et al., 2004). Here we will emphasize the importance of surface
elevation on PWV due to the high desert elevation of the Socorro measurement site. As most previous studies have focused on
lower surface elevations and tropical environments, there remains a need to easily determine the PWV in high elevation arid
climate zones for improved forecasting and trend monitoring.

There currently exist several methods for directly and indirectly measuring the total amount of water vapor in the atmo-
sphere. The more traditional methods of measurement include: Radiosondes (Guan et al., 2019; Li et al., 2003), ground-based
Global Position System signal delay analysis (Means and Cayan, 2013; Bevis et al., 1994), Solar Photometry (Raj et al., 2004;
Thome et al., 1992; Thomason, 1985), and Microwave Radiometry (Liljegren, 1994; Hogg et al., 1983). Though each method
has proven successful, the radiosonde remains the most widely used method to obtain atmospheric data. Some of the limi-
tations of using radiosondes to study the atmosphere include the cost of balloons and sensors, availability of personnel and
launch sites, and the frequency of launches.

Global Positioning Systems utilize a signal that passes through the atmosphere from a satellite to a ground-based receiver,
which measures the delay as a result of the amount of water vapor along the atmospheric path between the receiver and the
satellite (Means and Cayan, 2013). This signal delay can be used to estimate PWV assuming spatial homogeneity in conversion
to a vertical column of air. Many measurement sites are located near major airports and therefore typically do not supply rural
PWV measurements. Solar photometric methods apply a Langley extrapolation of multi-channel radiometric data in order to
quantify PWV. Sun photometers utilize both the 940 nm and 1020 nm near-infrared bands to determine PWV (Raj et al., 2004).

Relative to radiosonde data, this collection technique records PWV data with a precision of about 10%. (Thome et al., 1992; Thomason, 1985). Microwave Radiometers use Gigahertz frequencies to measure the incident microwave energy in the atmosphere, wherein the 23.8 GHz frequency is three times more sensitive to the concentration of atmospheric water vapor relative to the 31.4 GHz frequency (Liljegren, 1994). This two-channel approach enables a comprehensive profile of tropospheric water vapor and liquid water (Hogg et al., 1983). Limitations to using microwave radiometers to measure PWV include interference noise. There are also additional techniques that have been developed such as direct retrieval from ground-based hyperspectral IR observations (Turner, 2005), or calculated from thermodynamic profiles retrieved from hyperspectral IR observations (Turner and Blumberg, 2019).

Building upon a method using low-cost materials (under $50 USD) to determine PWV based on infrared temperature measurements of the zenith sky (Mims et al., 2011), we examined whether similar techniques could also be applied for higher elevation, arid and semiarid regions. A better understanding of this methodology may also demonstrate the feasibility of a citizen observer network, which could supply temperature data that would help monitor the PWV variations across different locations in a region. One major difference between our paper and previous work is our interpretation and modeling to better characterize and understand reasons for the correlation between zenith sky temperatures and PWV. Mims et al. first established the feasibility of this measurement technique, but their work was focused on observational results and provided little analytical interpretation. In addition, our measurement suite includes corresponding ground temperature data for instrument calibration and drift. For the remainder of the paper, we will discuss the observational methods including sensors and derivation of PWV from radiosondes (Sect. 2), results and analysis (Sect. 3), and interpretation using model simulations (Sect. 4). Conclusions are presented in Sect. 5.

## 2   Observational methods

We utilize infrared thermometry to measure the zenith sky (vertically upward at zenith angle of zero) temperature with a temporal resolution of approximately one day over Socorro, New Mexico (34N, 107W, 1.4 km surface elevation) (Kelsey and Riley, 2021) for a period of two years ($N_{\text{clear}}$ = 539). Three different handheld thermal sensors were used in this study: TE 1610[1], FLIR i3, and AMES.

### 2.1   Sensors

The FLIR i3 sensor has a hardware-imposed temperature measurement range from -20°C to 250°C (however, in our observations this sensor has produced temperature readings down to -40°C). The manufacturer defines the accuracy of these measurements as ±2°C and defines the spectral sensitivity to be between 7.5 and 13 $\mu$m (FLIR Systems Inc., 2012). Compared to the 4.8° conical field of view associated with the TE 1610 and AMES thermometers, the FLIR i3 has a 12.5°×12.5° rectangular field of view. The target emissivity for the FLIR is adjustable, but was set at 0.95 for consistency with the fixed value

---

[1]As a result of the lack of viable data from the TE 1610 (2 measurements out of 539 days), we have removed this sensor from further analysis and comparison.

employed by the AMES sensor.

The AMES thermometer has a low temperature measurement limit of -50°C and an upper limit of 550°C, and an assumed target emissivity of 0.95. The manufacturer error associated with this instrument differs between two temperature ranges. The first is between -50°C and 0°C, with an error of ±3.9°C (Harbor Freight Tools, 2017). The second range is 0.5°C and 550°C, the error for this range is ±2.2°C (Harbor Freight Tools, 2017). In our measurements we have found few instances where the temperature reading is below the -50°C threshold. There is not a defined spectral range provided by the manufacturer for this sensor. However, we inferred that the spectral sensitivity of the sensor lies within the range 7$\mu$m - 10$\mu$m by comparing to radiative model calculations, as further discussed in Sect. 4. We employed two sensors of this type: AMES 1, which was used starting on 22 January 2019, and AMES 2 which was put into service on 14 May 2019.

## 2.2 Measurement procedure

As discussed previously, the zenith sky temperature measurements are taken once a day, typically near 1700-1800 UTC or 2300-2400 UTC to avoid having the sun within the field of view of the sensors. Sky temperature is measured at the zenith, by hand, to facilitate the measurement of the vertical column air temperatures. This ensures the shortest optical path is used for infrared water vapor measurements (Smith and Toumi, 2008). A series of measurements were taken to investigate the impact of manual observations having small offsets from true zenith (determined by plumb bob, level, and large protractor), where readings were taken at varying angles up to 30° from zenith over a week-long period. It was determined that with proper technique one can manually get within 5° of true zenith, which introduces less than 0.8°C variation in clear sky temperature measurements. Angular variations in sky temperature might be expected to differ at other locations with different atmospheric conditions. We also measure the immediate ground temperature (the effective IR skin temperature) as a check on instrument calibration and drift. A noticeable difference in ground temperature measurements from one sensor in comparison to the others would highlight the need to look closer at the measurements from that particular sensor. A large variation in zenith sky temperature observations without a similar variance in the ground temperatures would flag that there may be an issue with the sky temperature measurement.

The presence of clouds, smoke, dust, or aerosol within the sensor field of view can have an impact on observed sky temperatures. Clouds, in particular, are capable of biasing the observations by providing an effective emission source at temperatures near cloud base. We screen and exclude any observations contaminated by clouds, regardless of cloud base height. This cloud screening is based upon subjective visual observations and Table 1 shows the breakdown of sky conditions and sensor readings for the entire data record. We find that cloud screening results in the loss of data for 26.5% of daily readings. Additionally, there are occasions when a given sensor will not produce a reading (NaN) when the sky temperature falls below the calibrated range for that sensor. This occurs mostly under clear skies, and it varies from 0.9% of the measurements for AMES 2 to 68.1% for FLIR i3 (Table 1). The larger fraction of NaN days for the FLIR i3 instrument is likely due to a warmer low-temperature cutoff (-40°C for FLIR i3 versus -50°C for AMES), and a different spectral sensitivity that is closer to the transparent atmospheric window between 8 and 12 $\mu$m wavelength. We have not made any measurements in the presence of noticeable smoke or dust. Surface solar radiation measurements at Socorro have shown that aerosol optical depths (AOD) are typically very low,

**Table 1.** Relative distributions of data types for Real, Not-a-Number (NaN), and No Data readings on the three sensors used in this study, broken down between clear sky and overcast labels. The overall fractions of measurements classified as either clear or overcast sky conditions are 73.5% and 26.5%, respectively.

| Sensor Label | Clear | | | Overcast | | |
|---|---|---|---|---|---|---|
| | % Real | % NaN | % No Data | % Real | % NaN | % No Data |
| FLIR i3 | 31.9 | 68.1 | 0.0 | 90.7 | 7.2 | 2.1 |
| AMES 1 | 96.1 | 3.9 | 0.0 | 96.4 | 1.5 | 2.1 |
| AMES 2 | 71.3 | 0.9 | 27.8 | 74.7 | 0.0 | 25.3 |

No Data is used for those days where data was not collected by the sensor.

varying between 0.03 and 0.10 with maximum values during summer (Minschwaner et al., 2002). These values are confirmed by sun-photometer data from the Sevilleta AERONET (AErosol RObotic NETwork) site located about 30 km north of Socorro (Holben et al., 1998, 2001). This AERONET site is near the Rio Salado riverbed and could be influenced by wind-blown dust, but despite isolated instances of high aerosol loading from either dust or wildfire smoke, AOD levels are typically no larger than 0.15. Variations in aerosol are not considered here, but they will contribute a small additional source of variability in sky temperature readings.

### 2.3 Infrared opacity and instrument comparison

As discussed above, the spectral sensitivity curves for each of our thermal sensors are not precisely known, but they are all assumed to have passbands that fall within relatively transparent atmospheric windows at wavelengths between $\sim$7 to $\sim$12 $\mu$m, corresponding roughly to the mid-IR spectral range. The downward mid-IR radiance observed at ground level with clear skies is primarily dependent on the vertical distribution of atmospheric temperature, and on the vertical distributions of greenhouse gases with mid-IR absorption signatures (e.g. Thomas et al. (1999)). The most important infrared-active gases at these wavelengths are ozone, with a vibrational band at 9.6 $\mu$m, and water vapor, with a weak continuum between the 6.3 $\mu$m vibrational band and the far-IR rotational lines of $H_2O$ (Stephens, 1994). Although the 9.6 $\mu$m ozone feature is significant for transmission through the entire atmosphere, most of the ozone is located in the stratosphere and ozone generally has a negligible impact on mid-IR transmission for path lengths within the lowest few kilometers of the surface, except perhaps under highly polluted conditions. On the other hand, even though the $H_2O$ continuum absorption is considered weak (only 10%-20% absorption through the entire atmosphere), the radiative effects can be significant for path lengths near the surface. The magnitude of this so-called e-type absorption varies as the square of the absorber amount (e.g. Burroughs (1979)). Furthermore, the scale height for the vertical distribution of water vapor ($\sim$3 km) is much smaller than for the background atmosphere, so that most of the water vapor continuum effects are felt within the lowest few kilometers of the surface.

Figure 1 shows instrument comparisons for clear sky temperatures and for ground temperatures, where the AMES 1 instrument is used as a standard due to its longevity and stability during the course of observations. We find that the AMES 1 and

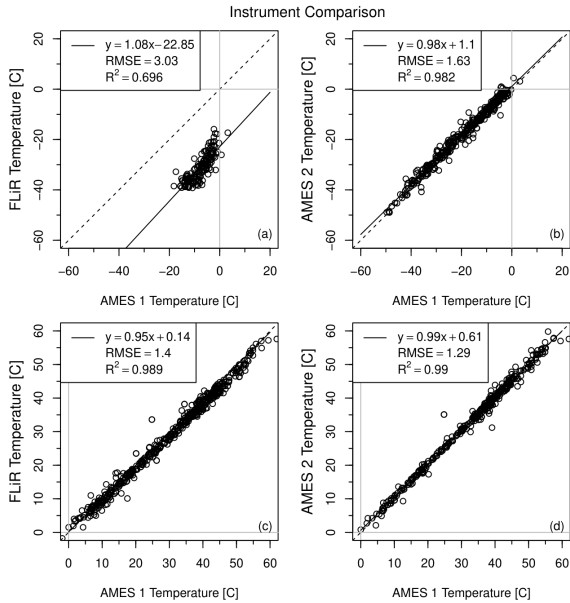

**Figure 1.** Comparisons between the AMES 1 and the FLIR i3 (left column) and the AMES 2 (right column) for clear sky (top row) and ground (bottom row). A 1:1 line is indicated as a dashed black line with the linear least-squares fit represented as a solid black line.

AMES 2 instruments agree to within $\pm\ 2°$C for both ground and sky temperatures, with no clear bias or offset. The FLIR i3 and AMES 1 instruments are also in good agreement for ground temperature, but they show a considerable difference in sky temperature. The FLIR i3 sensor consistently obtains readings that are $\sim20°$C lower than AMES 1 temperatures, and the difference grows larger with decreasing temperature. We believe that these differences are largely due to the difference in spectral passbands between the FLIR i3 and AMES sensors, with the FLIR i3 passband lying closer to the 8-12 $\mu$m atmospheric window, where water vapor opacity is a minimum and the effective emission occurs at higher altitudes and cooler temperatures, as shown in Appendix A. For these reasons and because of the differences in low-temperature cutoffs between the FLIR i3 compared with the other two sensors, the FLIR dataset is not included in our analysis.

## 2.4 PWV determination

There are no routine precipitable water measurements at Socorro, but two measurement sites are located within a 50-km radius: a PWV SuomiNet PWV ground station (Ware et al., 2000), and a sun-photometer installation within AERONET (Holben et al., 1998). In addition, there are two National Weather Service (NWS) stations within a few hundred kilometers of Socorro. Primarily due to data coverage issues, we adopted the NWS radiosonde PWV values for use in our analysis. The Socorro SuomiNet site is located only 2 km west of NMT campus where zenith sky temperatures are measured, but there are two reasons why the SuomiNet data are not used. First, this dataset has critical gaps in time coverage during our observing period - most notably over January-April and June-August of 2019. Second, the Socorro SuomiNet site is located on South Knoll, M-Mountain at

an elevation of 2.15 km above sea level, which is roughly 750 m higher in elevation than our measurement site. Assuming a water vapor scale height of 3 km, the difference in elevation could lead to a dry bias of 20% at South Knoll as compared to NMT campus. The AERONET site includes an automated sun photometer station on the Sevilleta Wildlife refuge, located approximately 30 km north of Socorro, but there is a significant data gap in the AERONET Sevilleta data from June 2019 to June 2020 that precludes the use of this dataset for our analysis. There is also a documented dry bias of 5-6% in AERONET sun-photometer PWV in comparison to radiosonde PWV (Pérez-Ramírez et al., 2014).

Our method to determine PWV at Socorro uses a weighted mean of balloon soundings from the NWS monitoring stations in both Albuquerque (ABQ) and Santa Teresa (EPZ). The ABQ NWS station is located approximately 110 km to the north of Socorro, while EPZ is located about 240 km to the south. The geographic locations Socorro, ABQ, EPZ, and topography of the region are shown in Fig. 2, along with the locations of the Socorro SuomiNet and Sevilleta AERONET sites. It should be noted that the elevation of the EPZ station is approximately 250 m lower than Socorro, while the ABQ station is approximately 200 m higher than Socorro. These differences are much less than the elevation difference between Socorro and the SuomiNet mountaintop site. Soundings from each NWS station are initiated at 0000 and 1200 UTC, which approximately brackets the ~1800 UTC Socorro temperature measurements; therefore, we average the 0000 and 1200 UTC soundings from each station to obtain daily means.

Figure 3 compares the radiosonde PWV data with both SuomiNet and AERONET observations for one year (2020). There is generally a good correspondence between ABQ and EPZ daily-mean PWV, consistent with previous studies that show spatial scales for PWV variations on the order of tens to hundreds of kilometers (Randel et al., 1996). However, larger differences between these two stations can be observed during periods when sharp gradients in humidity exist over central/southern New Mexico. In the following analyses, we employ a 2-dimensional interpolation (which is linear in horizontal distance, and exponential in elevation) of ABQ and EPZ radiosonde as a means to estimate PWV over Socorro. The corresponding linearized weighting factors are 0.75 for ABQ and 0.25 for EPZ.

During periods when PWV data from SuomiNet, AERONET, and radiosonde means are all available, the agreement shown in Fig. 3 is good. However, we do note a significant dry bias in SuomiNet compared with the radiosonde data, which is consistent with the difference in elevation discussed above. The amount of SuomiNet or AERONET data available for our two-year analysis period is even more limited than suggested for the single year shown in Fig. 3: out of 522 days with clear sky temperature measurements in 2019-2020, there are only 142 days of AERONET Sevilleta PWV data and 270 days of Socorro SuomiNet PWV data. However, these two datasets are still valuable comparison standards as discussed further below.

## 3 Results and analysis

### 3.1 Time series

A time series of two years of daily clear sky temperature and precipitable water is shown in Fig. 4. Both quantities show large seasonal cycles with maximum temperatures and PWV during the late summer and early fall (July-September). The seasonal amplitude in PWV is very large, with values averaging 5 mm during December-February and peaking at 25-30 mm in August.

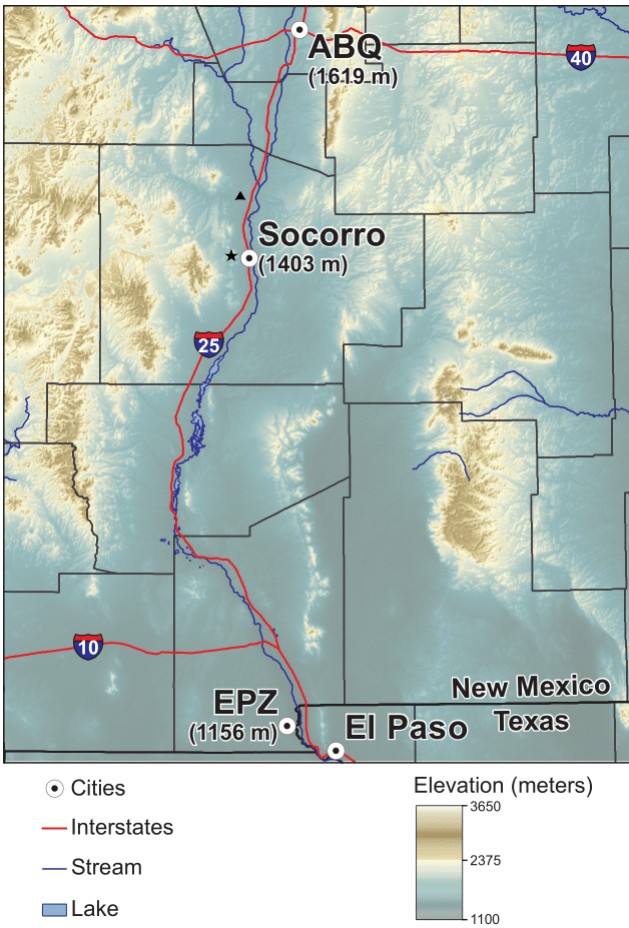

**Figure 2.** Topographical map of south-central New Mexico that shows the location of the measurement site at Socorro, along with the NWS radiosonde sites in Albuquerque (ABQ) and Santa Teresa (EPZ). The labels include surface elevations for all three locations. The black star and triangle indicate the locations of the Socorro SuomiNet and AERONET Sevilleta stations, respectively. Map prepared by Phil L. Miller, Map Production Coordinator, New Mexico Bureau of Geology and Mineral Resources.

This pattern is consistent with the timing of the North American Monsoon in New Mexico. The corresponding zenith sky temperatures range from -40°C in winter to -10°C in late summer. The day-to-day variability is on the order of 2-5 mm for PWV and roughly 2-5°C for temperature. Note the difference in PWV between Spring-Summer 2019 and Spring-Summer 2020, which provides some measure of the interannual variability. A detailed analysis of seasonal and interannual variability is beyond the scope of this paper; however, Appendix A presents evidence indicating that some of these differences are related to large scale changes in relative humidity.

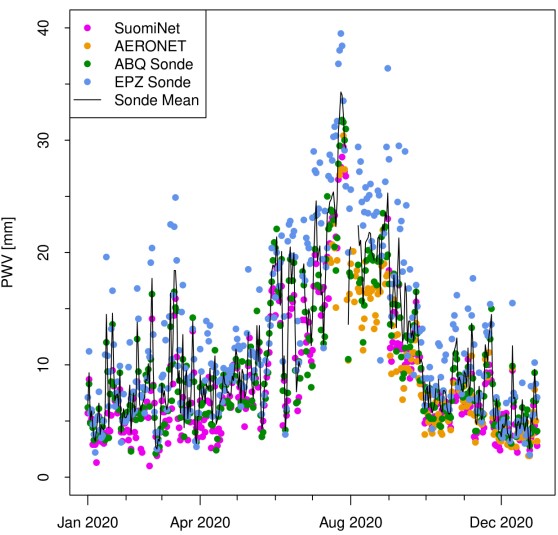

**Figure 3.** Time series plot showing daily mean SuomiNet, AERONET, and ABQ and EPZ radiosonde observations of PWV for the 2020 year (color symbols), with the weighted mean of the radiosonde measurements indicated by the solid line.

## 3.2 Analytical techniques

For the purpose of this experiment we developed the Precipitable-Water Model Analysis Tool (PMAT), which is a computational utility to analyze and visualize the collected data. Some of the visualizations used in the model include temperature and PWV measurements (as a function of time), direct sky temperature and PWV comparisons, and sensor performance comparisons. The tool implements common numerical methods to study the exponential relationship between the collected zenith sky temperature and PWV with ease (Riley and Kelsey, 2021). In the development of this computational model, we applied two common numerical methods: linearization of an exponential and least-square linear regression (LSLR).

We begin the process of analyzing the collected data by purging data that is not viable; this includes out-of-range temperature readings in addition to incomplete precipitable water measurements. Sensor malfunctions on radiosondes contribute to the missing precipitable water measurements. As part of this procedure, four additional days were not included in the final analysis because the results from these days exceeded a $3\sigma$ limit of deviation from the rest of the entire dataset.

For a more rigorous analytical process, we have implemented three additional pre-processing functions: a superficial overcast filter, a standard deviation filter with respect to the PWV observations, and a training-testing data partition. The first of the three simply removes data that has the overcast label. The second compares the standard deviation of the PWV observations for a given day with the mean of the daily standard deviations over the entire dataset, and rejects those days for which the standard deviation is more than twice the overall mean value. This filtering ensures that days with major differences between ABQ and

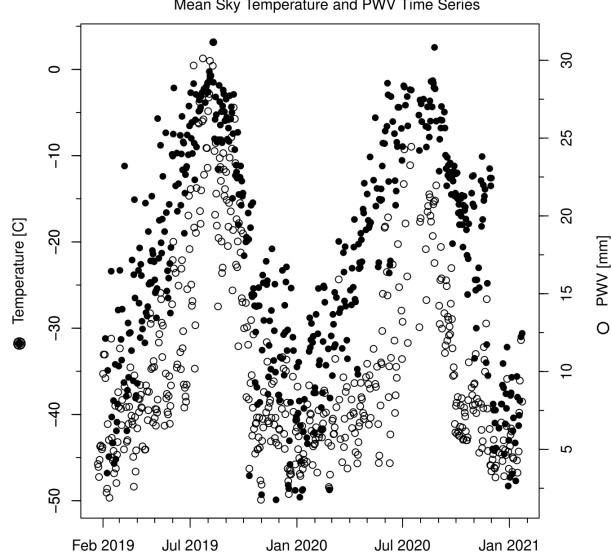

**Figure 4.** Time series composite of sky temperature (black) and precipitable water (white) over the period from January 2019 through January 2021.

EPZ PWV, or between 00Z and 12Z observations, do not negatively impact our analysis. Approximately 12% of the days are rejected by this filter. For the data partition, we split the data such that 80% was dedicated to training the regression model and the remaining 20% is for evaluating and testing the model.

After the data has been pre-processed, the relationship between the zenith sky temperature and precipitable water is passed through a least-squares linear regression algorithm in $(T_b, \log(\text{PWV}))$. For the purposes of this paper, we collected the parameters of the best-fit, the root mean squared error (RMSE), and the residual deviation ($S$) for the run. Then, we iterated the collection of the results for five thousand iterations. The results of the average best-fit curve is shown in Fig. 5. The exponential parameters for the best-fit function, physically defined as

$$\text{PWV} = Ae^{BT_b} \ , \tag{2}$$

as noted in Fig. 5, are $A = 18.48$ mm and $B = 0.034°\text{C}^{-1}$. The prediction interval, denoted as the shaded region, represents specific probability of future data points.

The goodness of fit in Fig. 5 confirms that a quasi-exponential relationship between the two variables provides a valid description of these observations. The scatter shown in Fig. 5 is dominated by the errors in PWV introduced by spatial/temporal averaging of radiosonde sounding from ABQ and EPZ, as discussed below. In view of the lower temperature threshold of -50°C, we see in Fig. 5 that the lowest detectable PWV using our technique is about 3 mm.

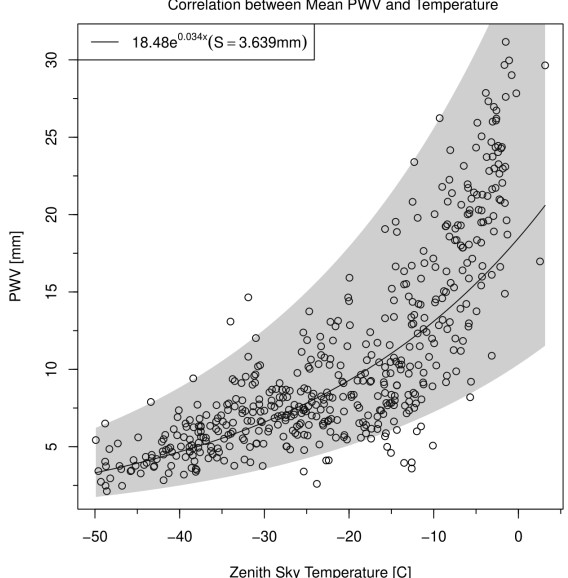

**Figure 5.** Analytical results of relationship between precipitable water vapor and zenith clear-sky temperature. Individual days are plotted with white circles. The solid black curve indicates an average best-fit exponential relationship based on five thousand iterations; this includes the 95% prediction interval of the data (shaded region). The fit parameters and residual standard deviation, $S$, are indicated in the legend.

## 3.3 Error analysis

The primary sources of error that impact our results include uncertainties in both the thermal sensors and in the radiosonde instruments. For the thermal sensors, the main sources of uncertainty are the precision of the instruments and uncertainties in zenith pointing. As discussed in Sect. 2.3, comparisons between sensors suggest that the precision is on the order of $\pm 2^\circ$C, while zenith pointing introduces uncertainties no larger than $0.8^\circ$C. Combined in quadrature, we obtain uncertainties in thermal readings of $\pm 2.2^\circ$C. For the radiosonde PWV uncertainties, we implemented the relative difference filter and applied a weighted mean of ABQ and EPZ to reduce errors from spatial-temporal averaging. However, based on the differences shown in Fig. 3 and taking into account known systematic biases due to elevation and differences between instruments (see Sect. 2.4), an uncertainty of 15% is estimated for daily mean PWV at Socorro as derived from the two radiosonde datasets.

To take into account perturbations in the sampled data generated by the standard deviation filter and data partition functions, we incorporated an iterative evaluation mechanism that collected five thousand steps worth of data. This process reduces the random bias that is included when we randomly partition the data set, so we can analyze a series of random states rather than a single state. In this evaluation, we calculated that the average residual standard deviation is 3.64 mm evaluated with the testing data subset.

In order to further gauge the accuracy and precision of our derived PWV, we compare the daily values of PWV based on our

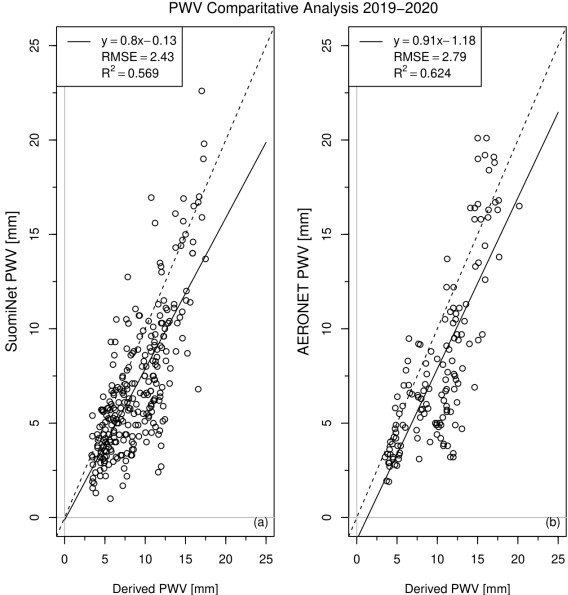

**Figure 6.** Comparison of SuomiNet PWV (left), and AERONET PWV (right) with PWV derived from our zenith sky temperature measurements using equation (2), over the period from January 2019 through January 2021. In both cases, the dashed line indicates a 1:1 relationship, while the solid line shows the linear least-squares fit, with parameters indicated in each legend box.

zenith sky temperature measurements, using the fit shown in Fig. 5, with all available 2019-2020 daily mean PWV values from SuomiNet and AERONET (Fig. 6). First, we note that there are more days available for comparison for SUOMINET (270) than for AeroNet (142). In both cases there are small variations which are due to differences in elevation, physical location, and instrument offsets. We find that SuomiNet PWV is generally lower (25%) than our derived PWV, which is consistent with the difference in elevation noted above. AERONET PWV also shows lower values of 15-20% with respect to our IR PWV product.

A large part of this apparent bias can be attributed to the differences in location and in instrument offsets as noted above.

The RMSE value calculated based on our model as a result of the aforementioned evaluations was, on average, 3.60 mm. For comparison, the RMSE generated from the evaluation of the Mims et al. best-fit with our dataset yielded a corresponding value of 4.63 mm. The lower RMSE for our model (∼20%) indicates that the relationship between PWV and clear sky temperature over Socorro, NM, is better described by our derived parameters rather than those that fit the more moist climate in eastern

Texas observed by Mims et al..

## 4  Interpretation and comparison to model simulations

In this section, we interpret the observed relationship between PWV and zenith sky temperature using radiative transfer calculations with the MODTRAN6 (MODerate resolution atmospheric TRANsmission 6) model (Berk et al., 2014). This framework

inputs vertical profiles of temperature, density, and radiatively active trace gases, and computes atmospheric spectral trans-
mittances and radiances over a wide spectral range from the ultraviolet through far infrared wavelengths. Here, we focus on
vertical path lengths through a midlatitude summer atmosphere (Anderson, 1986), with a surface located at 1.4 km altitude, in
order to simulate the zenith sky radiances at Socorro. In the base simulation, PWV = 11.4 mm and the temperature distribution
is unchanged from the midlatitude summer case. Other model runs include changes to the PWV by uniformly scaling the water
vapor vertical profile by factors of 0.5 and 2 while keeping temperature fixed, and by uniform temperature changes of -5 K and
+5 K while keeping PWV fixed.

Figure 7 shows downward spectral radiances computed within a wavelength range between 7 and 10 $\mu$m. This range is
taken for the sake of illustration because, as noted previously, the spectral passband of the AMES thermal sensor is expected
to approximately correspond to this region. The radiances shown in Fig. 7 can be used to separately quantify the impact of
changes in temperature or water vapor on downward radiances. We find that changes in PWV have the largest relative impact
on spectral radiances at 10 $\mu$m as compared to 7 $\mu$m, due largely to saturation effects closer to the edge of the strong $H_2O$ 6.7
$\mu$m band absorption. Changes in temperature, however, have a more uniform spectral effect.

For each case, we integrated the spectral radiances from 7 to 10 $\mu$m and determined the equivalent brightness tempera-
ture across this spectral range. The equivalent brightness temperature was found by integrating the Planck function over the
same spectral range, and solving for the blackbody temperature that provided the same integrated value as the MODTRAN6
downward radiances. The equivalent brightness temperature is intended to simulate our thermal sensor's zenith sky temper-
ature reading, and as indicated in Fig. 7, we do find a relationship between PWV and equivalent brightness temperature that
is somewhat consistent with the observations shown in Fig. 5. Higher PWV amount leads to higher effective temperatures,
which can be interpreted as a simple lowering in altitude of the effective emission level due to increasing opacity from water
vapor, and lower altitudes generally correspond to higher temperatures. For atmospheric temperature, we find an expected
increase/decrease in equivalent brightness temperature when atmospheric temperatures are respectively increased/decreased.

We developed a simple linearized model to further interpret our observations using the MODTRAN6 calculations above. If
the equivalent brightness temperature, $T_b$, is assumed to be a function primarily of PWV and atmospheric temperature, $T_{air}$,
then

$$\frac{dT_b}{d(\text{PWV})} = \frac{\partial T_b}{\partial(\text{PWV})} + \frac{\partial T_b}{\partial T_{air}} \cdot \frac{\partial T_{air}}{\partial(\text{PWV})} \ . \tag{3}$$

The observed relationship between $T_b$ and PWV is clearly nonlinear, but for small changes about some basic state ($T_b \simeq -20°$C
and PWV $\simeq 11$ mm) we assume that the observations can be represented by the left-hand side of Eq. (3) and that the slope
is approximately constant with a magnitude of about 1.9°C mm$^{-1}$ (Fig. 5). The MODTRAN6 simulations can be used to
estimate the partial derivative terms, so that the first term on the right-hand side of Eq. (3) has a magnitude of 1.04°C mm$^{-1}$
based on Fig. 7. This is the direct effect of changes in PWV on equivalent brightness temperature, and the results can be shown
to capture some, but not all, of the variations in the observed relationship. The second term on the right side of Eq. (3) accounts
for changes in $T_b$ that may arise from any coupling between $T_b$ and PWV due to changes in atmospheric temperature, and it
is composed of two factors. The first factor is 0.87 based on the MODTRAN6 calculations (Fig. 7). The second factor may

be estimated by assuming that the atmosphere maintains a state of constant relative humidity, so that the water vapor partial pressure at all levels (and hence PWV) is set by the Clausius–Clapeyron relation,

$$\frac{de_s}{dT_{\mathrm{air}}} = \frac{L_v}{R_v T_{\mathrm{air}}^2} \, , \tag{4}$$

where $e_s$ is the saturation vapor pressure, $L_v$ is the latent heat of vaporization, and $R_v$ is the specific gas constant for water vapor. If relative humidity is held fixed, then it can be shown that

$$\frac{\partial(\mathrm{PWV})}{\partial T_{\mathrm{air}}} = \frac{L_v}{R_v T_{\mathrm{air}}^2} \cdot \mathrm{PWV} \, . \tag{5}$$

Evaluating this equation for $T_{\mathrm{air}} = 273$ K and PWV=11.4 mm, we find the second factor on the far right side of Eq. (3) to be 1.21°C mm$^{-1}$, hence the entire second term has a magnitude of 1.05°C mm$^{-1}$. We conclude that the magnitudes of the two terms on the right side of Eq. (3) are nearly identical at about 1°C mm$^{-1}$, implying an overall slope on the order of 2°C mm$^{-1}$, which is in close agreement with the observed slope of 1.9°C mm$^{-1}$. A comparison of the model results to the observations is shown in Fig. 8. Despite the use of a simple linearized model to describe a clearly nonlinear relationship seen in the observations, we find a good level of agreement that confirms our hypothesis for the two primary influences on the relationship between PWV and zenith sky temperature.

In order to test the robustness of assumptions implicit in Eq. (5). we investigated the relationship between PWV and air temperatures near 3 km altitude using the Albuquerque sounding data spanning over one year. There was a considerable degree of scatter but PWV and air temperature were found to be well correlated, and a linear fit to the data (not shown) produced a slope consistent with the value estimated using Eq. (5). Figure 8 also includes the temperature-PWV relationship fit to observations by Mims et al. (2011), which employs an exponential form somewhat similar to ours. While the overall patterns are similar and consistent with the model, there are differences between the two fits that are most likely due to different sensitivities between the sensors used, and to differences between climate regimes (e.g., mean relative humidities for our location are much lower than for the Mims et al. (2011) study).

## 5   Conclusion and future directions

Our results demonstrate the feasibility of using low-cost sensors to measure PWV in less than five minutes using simple measurement protocols, confirming the findings by Mims et al. (2011), but our work extends the previous analysis by observing at colder zenith sky temperatures (down to -40°C) and correspondingly lower PWV (down to ∼3 mm). Our measurements also show that the exact $T_b$ - PWV relationship will be a function of instrument spectral sensitivity and local conditions such as surface elevation and mean relative humidity. In addition, we developed a simple model that uses MODTRAN6 radiative transfer calculations to quantify how $T_b$ can be influenced by changes in PWV and in mean-column air temperature, an analysis that was not done in previous studies. The model analysis indicates that the observed relationship between zenith sky temperature and PWV can be explained primarily by two dominant influences. First, an increase in PWV leads to increasing atmospheric opacity and a lower altitude for the effective emission height as viewed from the surface. Under typical conditions a lower

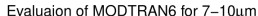

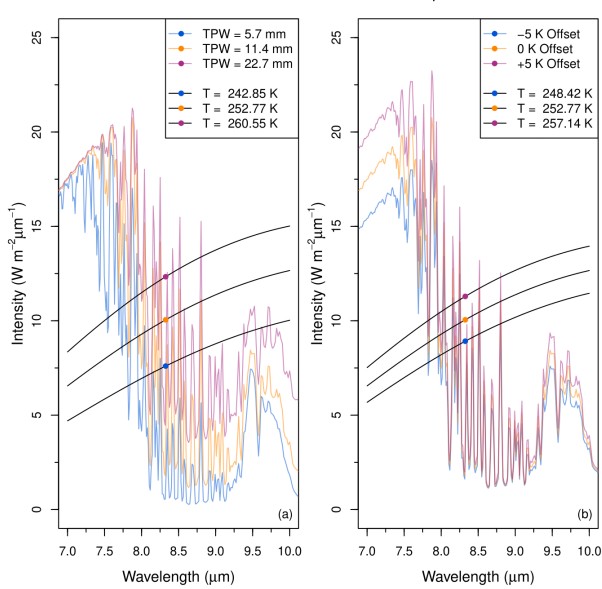

**Figure 7.** (Left) Downward radiances at the surface, located 1.4 km above sea level, computed using MODTRAN6 with a midlatitude summer model atmosphere. Water vapor was uniformly scaled to produce a PWV of 5.7 mm (blue), 11.4 mm (orange), and 22.7 mm (purple), with all other model parameters held constant. Also plotted are blackbody curves for three different temperatures (black), with each curve having the same integrated radiance as the corresponding MODTRAN6 radiance ($T_b$=260.55 K, 252.77 K, and 242.85 K for PWV=22.7 mm, 11.4 mm, and 5.7 mm, respectively). (Right) Same as Left but for uniform changes in atmospheric temperature of $\pm 5$ K with water vapor held constant at PWV=11.4 mm. Equivalent blackbody temperatures are $T_b$=257.14 K, 252.77 K, and 248.42 K for the case of $\Delta T_b$=+5 K, 0 K, and -5 K, respectively.

height corresponds to a higher temperature. Second, an increase in PWV is typically correlated with higher air temperatures; although relative humidity is not perfectly constant, the climatology is such that positive relationships between temperature and humidity are generally observed. Higher air temperatures, in turn, increase the observed zenith sky temperature due to greater emission rates governed by the Planck function, as seen in the MODTRAN6 simulations. The model results show that surface elevation and climatological relative humidity are two of the most important local factors in shaping the exact form of the $T_b$ - PWV relationship.

Since PWV can typically be measured to within $\pm 20\%$ using this approach with a single-design sensor, it shows promise for applications involving a dense network of PWV observations, and it may be a good candidate for broader observations employing the "citizen science" methodology. Coordinated observations within the Global Learning and Observations to Benefit the Environment (GLOBE) Program has been proven to be successful for a wide variety of geophysical phenomena (e.g., Robles et al. (2020)). The question of whether or not sensors of the same model and manufacturer are similar enough to be used in an observing network is an area of future investigation. We also found that those sensors which were not capable of measuring

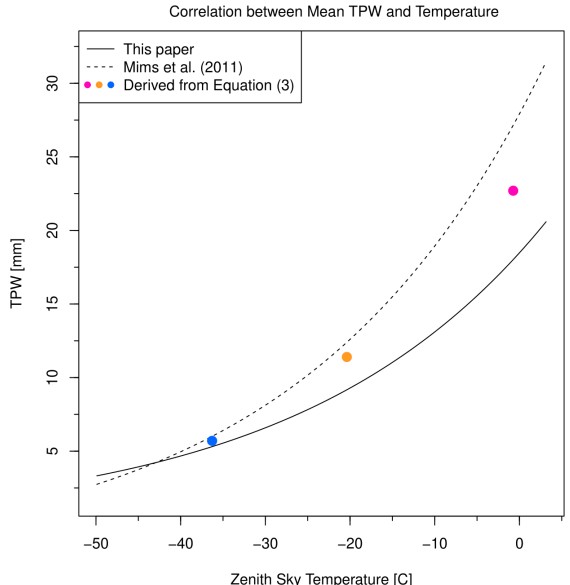

**Figure 8.** Zenith sky temperature versus PWV for the best fit to measurements from this study (solid curve), best fit from Mims et al. (2011) using the expression $30.55e^{0.035x} - 2.63$ (dashed curve). Results from MODTRAN6 radiance calculations for average 7-10 $\mu$m effective brightness temperature (blue, orange, and purple solid circles) are plotted for comparison. The model results include the combined effects of changes in PWV and air temperature on the effective brightness temperature, as expressed by Eq. (3).

temperatures colder than -20°C were not able to collect zenith sky temperature data in Socorro, New Mexico. However, at a lower elevation and less arid region, zenith sky temperatures rarely fall below -20°C [e.g. Mims et al. (2011)], and those sensors may be effectively utilized for PWV monitoring

As we continue the study of the relationship between zenith sky temperature and precipitable water, we plan on developing an autonomous sensor module. This module would not only enable consistent temperature measurement times, but will also facilitate an expansion of this project with more measurement sites. Additional measurement sites will increase our capability to analyze the relationship between zenith sky temperature and precipitable water in different climate zones. We are also developing plans to work with schools to continue manual data collection in different parts of the American West to help advance science learning while collecting data from regions with different elevations and precipitation profiles. Current efforts are focused on testing and optimizing a machine learning algorithm (more specifically a Support Vector Machine) to predict a binary set of weather conditions, clear sky or overcast, based on zenith sky temperature and PWV data. These predictive models will have the capabilities to further quantify the aforementioned relationship by applying common statistical metrics, and will be the subject of a future paper.

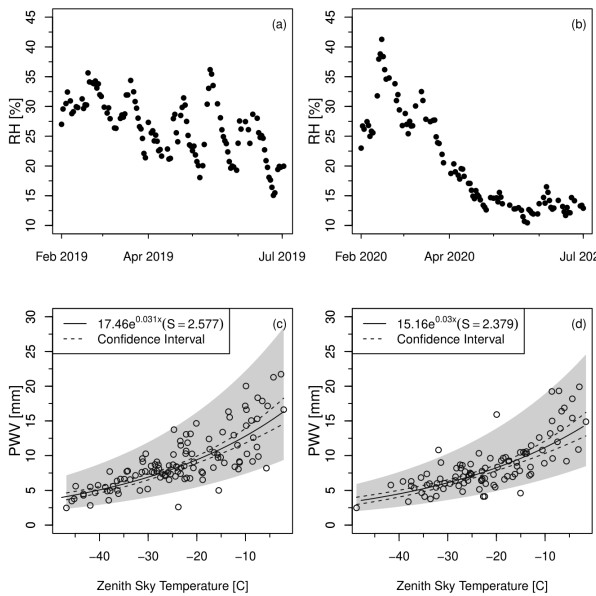

**Figure A1.** Top two plots show time series of surface relative humidity measured at Socorro, New Mexico. The left plot shows the period from February to July 2019, and the right plot shows the values for February-July 2020. The bottom two plots show the corresponding PWV and zenith sky temperatures for the same periods in 2019 (left) and 2020 (right).

## Appendix A

This Appendix presents two supplementary figures that support discussions about variability in PWV and spectral passbands of our instruments.

Figure A1 shows a comparison of surface relative humidity measured at Socorro, NM for the first halves of 2019 and 2020, along with the corresponding $T_b$ and PWV measurements analyzed over same two time periods. We find that relative humidity (RH) values in late spring and early summer of 2020 were much lower than those observed in 2019. Similarly, PWV values in
Spring-Summer 2019 were lower in 2020 compared with 2019. However, measured values of $T_b$ did not undergo a proportional change so that the 2019 and 2020 relationships show small differences that can be seen in the fits. The reductions in RH and PWV appear to be consistent with the La Niña pattern seen in 2020, although a more complete analysis would require more years of $T_b$ and PWV measurements.

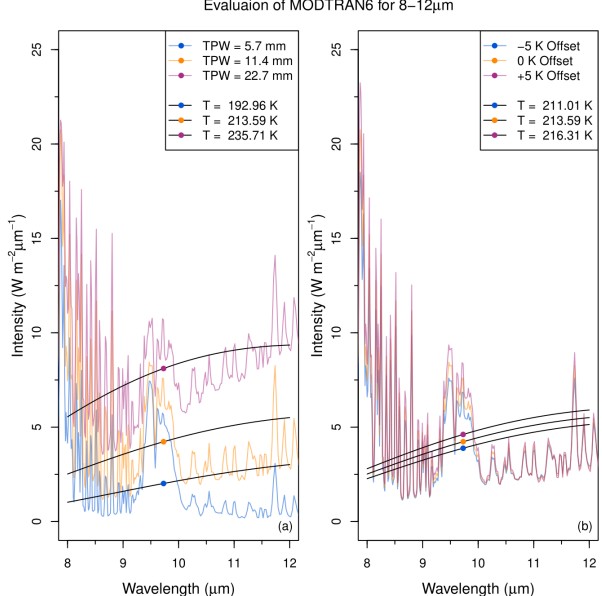

**Figure A2.** (Left) Downward radiances at the surface, between 8 and 12 $\mu$m, computed using MODTRAN6 with a midlatitude summer model atmosphere. Water vapor was uniformly scaled to produce a PWV of 5.7 mm (blue), 11.4 mm (orange), and 22.7 mm (purple), with all other model parameters held constant. Also plotted are blackbody curves for three different temperatures (black), with each curve having the same integrated radiance as the corresponding MODTRAN6 radiance. (Right) Same as Left but for uniform changes in atmospheric temperature of $\pm5$ K with water vapor held constant at PWV=11.4 mm. Equivalent blackbody curves and temperatures are also shown.

Figure A2 shows the results of MODTRAN6 calculations as described above in Sect. 4. In this case the spectral passband is assumed to be between 8 and 12 micrometers, and we find a corresponding decrease in effective brightness temperatures compared to those calculated for the 7-10 micrometer spectral region shown in Fig. 7. These results confirm our hypothesis that the lower temperatures observed by the FLIR i3 instruments are primarily due to differences in spectral passbands. Furthermore, we find a much reduced sensitivity to air temperature within this passband, suggesting that this kind of instrument could provide

a more direct means of monitoring PWV, but only for climate regimes where mean humidities are sufficiently large so that observed sky temperatures would fall within the measurement temperature range for this instrument.

*Code and data availability.* The data that support the findings of this study are available from https://doi.org/10.6084/m9.figshare.12712814. The model code that was used in this study are available from https://doi.org/10.5281/zenodo.4587475

*Author contributions.* Vicki Kelsey conducted most of the zenith sky temperature measurements and developed the methods used for estimating local PWV from NWS data. Spencer Riley developed the analysis codes and prepared all figures in the paper. Kenneth Minschwaner assisted with measurement logistics and developed the simple linearized model. All three authors contributed equally to writing the paper.

*Competing interests.* The authors declare that they have no conflict of interest.

*Acknowledgements.* The authors would like to acknowledge the New Mexico Tech Physics Department and the Langmuir Laboratory for Atmospheric Research for their technical support of this project. The authors especially wish to thank Sooraj Bhatia, Christopher Baca, Brandon Phelps-Romero, Steve White, Fernando Rivera, Damon Apps, Eloise Apps, and Charles Apps for assisting with zenith sky and ground temperature measurements. We thank both Dr. Nelia Dunbar and Phil Miller from the New Mexico Bureau of Geology and Mineral Resources for graphical assistance for Figure 2. We thank Dr. David Meier for useful discussions and acknowledge Altagracia Lujan from the New Mexico Tech Physics Department for her administrative support during this project.

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
