# Peer review of "Atmospheric Precipitable Water Vapor and its Correlation with Clear Sky Infrared Temperature Observations"

_Atmospheric Measurement Techniques, 2021_

## Author Comment (AC2)

We thank both reviewers for their feedback and recommendations for improving the manuscript. We have adjusted the paper to take into account the responses from both reviewers, and a point-by-point explanation of those changes is presented below.

**Reviewer 2**

**General Comments**

*The work entitled "Atmospheric Precipitable Water and its Correlation with Clear Sky Infrared Temperature Observations" investigates an indirect retrieval approach for Total Precipitable Water based on zenith sky temperature measurements using low-cost infrared thermometers. The approach used by the authors is interesting and scientifically accepted. I appreciate the general flow of the paper; the technique is described in detail and the topic is suitable for the scope of the journal.*
*I therefore recommend that the manuscript is accepted for publication after the following minor changes.*

**Specific Comments**

*A) Please further emphasize the novelty element of this work with respect to the previous related one, both at the end of the Introduction and in the Conclusions.*

> Reviewer #1 had a similar suggestion, and we have addressed this issue in both introduction and conclusion

*B) I am concerned about the TPW dataset, given the large distance between the measuring sites. Could you add any other source, closer to the site of interest?*

> We have investigated other measurement sites (SUOMINET and AERONET, please see response to reviewer #1) that utilize a variety of different measurement techniques and have found that the NWS radiosondes are the most reliable sources of data available. We have added a discussion in the manuscript and a figure in the appendix to further address this issue.

*C) Moreover, while I do not know if this is feasible and meaningful here, I reckon that separating your dataset into training/test subsets would be beneficial for this work, so that you could evaluate the fit on an independent dataset via the standard statistical analysis, hence improving the overall quality of this paper.*

> We have implemented an 80/20 training and testing split into our analysis. Our new figure 4 includes the best-fit line generated by the training set and our analysis looks

into the results of our testing set to evaluate the fit. We have updated the appropriate sections to reflect these changes.

*D) I wonder whether the –50 degrees instrument threshold has not been too strict a limit in this work and related measurements. Please add a few statements explaining why this has (not) been a limiting issue in your work.*

The -50 degrees threshold is a limitation built into the sensor itself, not something we have control over. The paper has been revised to clarify this point. Fortunately, we have not seen a significant number of AMES measurements exceed the instrument threshold so this has not been a major concern.

*E) Finally, I believe this is a paper on the retrieval of Integrated Water Vapor (IWV), since all measurements are in clear sky. If so, I would suggest rephrasing through the whole manuscript.*

Per the recommendation of reviewer#1, we have updated the paper to use the term Precipitable Water Vapor (PWV) rather than Total Precipitable Water (TPW) or IWV.

**Technical Corrections**

*Line 5: "We have analyzed this relationship: what relationship are we talking about? Please amend accordingly.*

This refers to the relationship between zenith clear sky temperature and PWV. The paper has been revised to clarify this point.

*Line 10/11: "but with parameters that are different than those obtained for the Gulf Coast". What are you referring to? Please add detail.*

We were referring to the North American Gulf Coast (Texas), the location of the Mims et al measurements. We have updated the document to explicitly state this.

*Line 36: I suggest replacing "Under clear skies that are the focus of our work" with "In clear sky (the focus of this work),"*

Change implemented in revised manuscript.

*Line 51: Please provide an adequate number of references for each method.*

We have added citations to this section of the manuscript.

*I suggest to completely remove the TE1610 sensor discussion from the paper, as I understand it has been no use for this work.*

Discussion of the TE1610 sensor has been removed.

*Line 174-175: I find this statement redundant, and overall, I am not expecting to see any type of results from the FLIR instruments, given that you decided not to include any. **Again, in line 195,** I guess there's no need to mention it. I suggest keeping only the discussion about instruments/dataset effectively used in the end, as this would make the work neater and improve its flow. However, the discussion in the appendix is just fine, as it "proves" the reason why FLIR3 was not used.*

Discussion of the FLIR has been condensed to reflect the reviewer's comments.

---

## Author Response (AR1)

All values reported in the revised manuscript are correct, we have addressed a few minor issues with the analysis that caused some quantities to be incorrectly reported in our original response to the reviewers.

| R# | Reviewer's Comment | Authors Response | Changes Made |
|---|---|---|---|
| 1 | The first thing that struck me while reading this paper is that this is not a method to observe total precipitable water (TPW), but really a method to observe precipitable water vapor (PWV) in clear sky conditions. While one can argue that in clear skies the TPW is functionally equivalent to the PWV since there is no liquid or ice water present, this distinction is a valuable one: there are more sources of PWV data than TPW since measuring cloud characteristics is so challenging. There are several additional ways of measuring PWV that the authors do not address in the manuscript. This includes a direct retrieval from ground-based hyperspectral IR observations (Turner 2005 https://doi.org/10.1175/JAM2208.1), calculated from thermodynamic profiles retrieved from hyperspectral IR observations (Turner and Blumberg 2018 https://doi.org/10.1109/JSTARS.2018.2874968 ), Raman lidar, aircraft, etc. | We have revised the paper to utilize the term Precipitable Water Vapor (PWV) in place of Total Precipitable Water (TPW). We have also included a brief discussion on the additional techniques that were recommended, including citations. | We have changed TPW to PWV throughout the paper.

Line 62 (revised): Added "There are also additional techniques that have been developed such as direct retrieval from ground-based hyperspectral IR observations \citep{Turner_2005}, calculated from thermodynamic profiles retrieved from hyperspectral IR observations \citep{Turner_2019}." |
| 1 | This leads into the most significant concern that I have about the present work: the training and validation dataset has significant drawbacks and better choices may be available. It may be true that in the desert southwest the temporal and spatial variability is not large, but it remains that the data being | We have investigated alternative data sources for PWV, including SUOMINET and AERONET. The reviewer specifically mentioned the Socorro SUOMINET site, and although we have leveraged this dataset for partial validation of our use of NWS radiosonde PWV data (now discussed in the manuscript and in | Line 120 (revised): Added "These values are confirmed by sun-photometer data from the Sevilleta AERONET (AErosol RObotic NETwork) site located about 30 km north of Socorro \citep{Holben_1998, Holben_2001}. This AERONET site is near the Rio Salado riverbed and could be |

used is, at a minimum, located 110 km and 6 h away from the desired quantity. I am surprised that the authors did not utilize the Suominet observations of PWV from the Socorro area, especially since one of the authors is the contact for that particular observing site. This may be due to thinking that the present work describes a TPW product and not a PWV product. It is true that the observation site is located on a mountain while the IR observations are presumably taken at a lower altitude. This criticism is tempered somewhat by the fact that the two radiosonde sites used for validation differ in elevation by ~400 m and so altitude differences are going to be an issue regardless of the validation set used. That being said, a quick glance at a 14 day time series at Albuquerque (http://www.atmo.arizona.edu/products/gps/P034_14day.gif) and Socorro (http://www.atmo.arizona.edu/products/gps/SC01_14day.gif) doesn't really show a huge impact of the altitude (at least at the time of the writing of this review). Suominet has the advantage of a substantially better temporal resolution allowing a more direct comparison to the IR observations, and in fact, offering enough observations that it would be possible to average to reduce noise in the signal.

the appendix), there are two reasons why these data have not been adopted in the analysis. First, the SUOMINET data set has critical gaps in time coverage - most notably over January-April and June-August of 2019. In addition, and as noted by the reviewer, the Socorro SUOMINET site is located on South Knoll, M-Mountain at an elevation of 2.15 km above sea level, which is roughly 750 m higher than NMT campus where the zenith sky temperatures are measured. This is a significant difference, and much larger than the difference in elevation between NMT campus and either the Albuquerque and El Paso NWS stations. Assuming a water vapor scale height of 3 km, this could lead to a ~20% systematic difference between South Knoll and NMT campus. Note that the elevation differences of ~200 m between NMT campus and either El Paso or Albuquerque are expected to lead to ~7% differences, and these are mitigated by the use of weighted averages from both sites. Complete details are now included in the revised manuscript.

In regard to AERONET, there is an automated sun photometer station located at the Sevilleta Wildlife refuge, located approximately 30 km north of NMT campus. We have also used PWV data from this site for validation purposes (also now discussed in the manuscript and appendix), but there is a significant data gap in the AERONET Sevilleta data from June 2019 to June 2020, which precludes the use of this dataset for our analysis. There is also a documented dry bias of 5-6% in AERONET sun-photometer PWV that must be considered

influenced by wind-blown dust, but despite isolated instances of high AOD from either dust or wildfire smoke, it is typically no larger than 0.15. Variations in aerosol are not considered here, but they will contribute a small additional source of variability in sky temperature readings."

Line 151-161 (revised): We have added an extensive discussion on both the AERONET and SuomiNet data sources.

Line 346-352 (revised): Added Appendix B with supporting information in comparisons with the AERONET, SuomiNet, and radiosonde data sources.

| | | | |
|---|---|---|---|
| | | (Perez-Ramirez et al., JGR, 2014). Overall however, our comparisons of SUOMINET, AERONET, and NWS radiosonde data over limited time periods have led to a refinement in averaging data from the two NWS sites, and to a better understanding of the limitations in using this technique to estimate PWV. | |
| 1 | Even if they choose not to use Suominet observations, there are ways that the radiosonde dataset can be leveraged to create a more representative data sample. Rather than using every single IR observation, it may be better to exclude from analysis the cases in which there is a substantial difference between the two sites, and/or between the 0000 and 1200 UTC launches. By focusing on cases in which the spatiotemporal variability is small, the authors can have greater confidence in the retrieved product. This will reduce the number of data points, but I feel will produce a stronger product overall. | In response to this feedback, we have investigated additional ways to address the issue of large spatiotemporal variability and small resolution. The result of our research is the implementation of a weighted average on the PWV data that better reflects the distances between the two NWS sites and Socorro, NM. We have included a discussion on this process in the analysis section. In addition, we have implemented a data screening function that excludes PWV data for which the difference between the two sites is larger than 75% of the unweighted mean. This threshold was defined so that no more than 10% of the complete dataset is excluded. | Line 201 (revised): Added "The second compares individual PWV observations to the daily mean of both ABQ and EPZ, and rejects those days for which any difference exceeds a fixed threshold of 55%. This threshold value was determined so that no more than 10% of the days are rejected by this filter, while still ensuring that days with major differences between ABQ and EPZ radiosondes do not bias our analysis"

Line 191 (markup): Changed "an unweighted mean" to "a weighted mean (inversely related to distance from Socorro)" |
| 1 | The error analysis also seems to be somewhat lacking, as it tends to focus on the uncertainty of the regression while not addressing the influence of the uncertainty of the instrument or the measurement technique. A monte carlo approach may prove useful here: by randomly perturbing the input brightness temperatures by a random value chosen from a gaussian distribution with a standard deviation equal to the instrument uncertainty, then repeating that | Thank you for the feedback. We have developed and explored a few additional analysis techniques that have been added to the paper. The first is a testing/validation data partition mechanism with an 80/20 split. We have also recorded more relevant metrics for gauging the dataset and the regression analysis. The revised paper now includes a discussion of this method.
While we have not implemented a Monte Carlo | Line 204 (revised): Added "For the data partition, we split the data such that 80% was dedicated to training the regression model and the remaining 20% is for evaluating and testing the model."

Line 207 (revised): Added "For the purposes of this paper, we collected the parameters of the best-fit, the root mean squared error (RMSE), and the residual |

| | | | |
|---|---|---|---|
| | over a set number of trials, it may provide a more realistic assessment of how the instrument itself may be contributing to the error bars of the retrieved value. This doesn't include the uncertainty induced by the way the instrument is held, which may also expand the uncertainty of the retrieved value. | approach for this paper, we are looking at developing this as a part of future analysis. The quantity of data is insufficient to justify a full Monte Carlo analysis at this time. | deviation (S) for the run. Then, we iterated the collection of the results for five thousand iterations."

Line 218-234 (revised): Reworked Error Analysis subsection to reflect changes. |
| 1 | Finally, I'd like to see a greater exploration of the differences between Mims et al 2011 and the present work. What is the RMSE of the current dataset, and how does that compare to the RMSE if you applied the Mims relationship to your data? In other words, how much are you improving the technique by tuning it for your specific location? Such an analysis would help increase the novelty of this paper. | One major difference between our paper and Mims et al 2011 is our interpretation and modeling to better characterize and understand reasons for the correlation between zenith sky temperatures and PWV. The Mims et al 2011 paper included no such analysis and focused strictly on the observational results. In addition, our measurement suite includes corresponding ground temperature data for instrument calibration and drift, which was not discussed by Mims et al. These points are now emphasized more heavily in the paper.
We thank the reviewer for the suggestion of further comparison with Mims et al. As a part of our revised analysis section, we have explored the comparison between the Mims et al, 2011 fit and our fit for the Socorro measurements.. We found that the RSME associated with the Mims et al fit was 4.52 mm while the corresponding value for our fit is 3.82 mm. This is a significant enough change to warrant the "tuning" of this technique to our specific location. Also note that these values are not filtered, with the exception of the overcast filter, and includes all of the clear sky measurements. | Line 69 (revised): Added "One major difference between our paper and previous work is our interpretation and modeling to better characterize and understand reasons for the correlation between zenith sky temperatures and PWV. Mims et al. first established the feasibility of this measurement technique, but their work was focused on observational results and provided little analytical interpretation. In addition, our measurement suite includes corresponding ground temperature data for instrument calibration and drift."

Line 230-234 (revised): Added comparison between the RMSE values of the Mims best-fit and our results. |

| 1 | Line 50. Consider how PWV (not TPW) is also being measured by various systems, based on the discussion above. | Please see our response above to the first Specific Comment. | We have changed TPW to PWV throughout the paper. |
|---|---|---|---|
| 1 | Line 75. How are the observations actually being taken? Is a human pointing a hand-held system towards the sky and writing down the observed temperature, or is a more robust method being used? Many IR thermometers have adjustable emissivities, and the default isn't necessarily a blackbody. Were the emissivities set to the same value across all systems? | The measurements were taken by a human pointing the hand-held device at the zenith sky. While many IR thermometers have adjustable emissivities, the thermometers we employed in this research had constant emissivities of 0.95. The paper has been revised to include this information. | Line 86 (revised): Added "The target emissivity for the FLIR is adjustable, but was set at 0.95 for consistency with the fixed value"

Line 88 (revised): Added "and an assumed target emissivity of 0.95."

Line 108 (revised): Added "by hand" |
| 1 | Line 77. Does the manufacturer note the wavelengths at which this instrument operates? | We were able to locate the particular technical manual that states that the TE 1610 has a spectral response of 8 - 14 micrometers. However, the paragraph discussing the TE 1610 was removed per the recommendation of reviewer #2. | The section regarding the TE 1610 was removed from the revised paper. |
| 1 | Line 99. This analysis of how to hand-hold a thermometer within 5 deg of zenith, and the fact that it results in less than 1 C uncertainty, is interesting, and the discussion of both points should be expanded. | Through the utilization of a protractor and level, we have verified that a trained observer can consistently point a hand-held sensor to within 5 degrees of zenith. Using the same setup we also mapped the distribution of temperature versus zenith angle. The typical changes in temperature over a 5o cone centered on zenith are no more than 0.8oC. This is now discussed in the paper. | Line 104 (revised): Added "Angular variations in sky temperature might be expected to differ at other locations with different atmospheric conditions"

Line 101 (revised): Added "(determined by plumb bob, level, and large protractor),"

Line 103 (revised): Changed +1 C to 0.8 C |
| 1 | Line 104. How are you screening for clouds? Observer judgement? Airport ceilometer? Satellite? IR thermometer threshold? | The current method of classifying the dataset is based on observer judgement. Early into the project we considered an IR temperature | Line 111 (revised): Added "This cloud screening is based upon visual observations..." |

| | | threshold, but found that this method was inconsistent with visual observations due to variations in cloud base altitudes over Socorro. The paper has been revised to clarify this further. | |
|---|---|---|---|
| 1 | Line 111. I find it surprising that there is little dust in the middle of the high deserts of New Mexico. Why is the dust so low? | Wind-blown dust can be a problem in certain areas of New Mexico, but Socorro is located in the Middle Rio Grande Valley and does not experience widespread dust episodes. Isolated areas of dry creek beds can, however, be affected during high wind episodes in the spring season. As noted in the paper, "Surface solar radiation measurements at Socorro have shown that aerosol optical depths are typically very low, varying between 0.03 and 0.10 with maximum values during summer (Minschwaner_2002)." We verified this using the sun-photometer data from the Sevilleta AERONET site located about 30 km north of Socorro, which is also near the Rio Salado riverbed and should be even more influenced by wind-blown dust. Despite isolated instances of high AOD from either dust or wildfire smoke, AOD is typically no larger than 0.15. We have included a sentence with the additional AERONET analysis in the revised paper. | Line 118-124 (revised): Added a discussion regarding the aerosol optical depth and the impact of aerosols on our observations. |
| 1 | Fig 1. This figure is very confusing to me, and I apologize if there is something obvious that I'm missing. There are four categories: clear, cloudy, clear NaN, cloudy NaN. It seems like two separate things are going on. There is an instrument assessment to determine if the sky is clear or not (more detail on that is needed). But in the case of the NaNs, an external | In place of Figure 1, we have developed a table to clear up some confusion. The table states the percentage of clear sky days out of the total number of data points, and then the percent of NaN values out of the clear sky. From this feedback we have also drafted new designs for a replacement figure in the software. | Fig 1: Changed to Table 1 that more clearly indicates the data type distribution. |

| | | | |
|---|---|---|---|
| | assessment of the clear our cloudy state has to be used because the instrument is not reporting anything. This is all coupled with the fact that the manuscript says that clouds were filtered out. Ultimately, I'm not sure what the figure is trying to tell me. A better approach may be a contingency table for each instrument that compares the external / instrument assessment in terms of clear/clear, clear/cloudy, cloudy/clear, and cloudy/cloudy, with special notes of the number of NaNs in each category. | | |
| 1 | Figure 2. By starting out the caption with (a,c) it is somewhat confusing to the reader (who may be more accustomed to going from a to b). It may be better to say something like "Comparisons between the AMES 1 and the FLIR i3 (left column) and the AMES 2 (right column) for clear sky (top row) and ground (bottom row)." | We thank the reviewer for this suggestion to improve clarity and have made appropriate revisions. | Figure 1 (revised): Caption was changed to be "Comparisons between the AMES 1 and the FLIR i3 (left column) and the AMES 2 (right column) for clear sky (top row) and ground (bottom row). A 1:1 line is indicated as a dashed black line with the linear least-squares fit represented as a solid black line." |
| 1 | Line 140. This section would be greatly improved with a map showing the location of ABQ, EPZ, and Socorro, with elevation as the background color. | We appreciate this suggestion and a map has now been included to show locations and elevations of the region of interest. | Figure 2 (revised): A topographical map of the region of interest has been added.

Line 362 (revised): Added "We thank both Dr. Nelia Dunbar and Phil Miller from the New Mexico Bureau of Geology and Mineral Resources for graphical assistance for Figure 2." |
| 1 | Line 156. The amount of data that is used in the analysis fits better in the methodology than in the results. I found myself using the values | We have updated the paper such that the amount of data is now recorded in the methodology. | Line 79 (revised): Added "for a period of two years ($N_{\text{clear}}$ = 539)" |

| | | | |
|---|---|---|---|
| | reported in Fig 1 to calculate the approximate number of datapoints for context before I got to this part of the paper. | | |
| 1 | Line 186. Is this R^2 for a linear correlation? If so, you may actually have a better fit than your numbers report, since the fit has an obvious non-linear shape. | We have updated the figure and the discussion to report the residual standard deviation rather than the coefficient of determination (R^2). | Figure 4 (revised): Replaced R^2 value with the residual standard deviation (S).

Line 239 (markup): Removed "coefficient of determination (R^2) associated with this relationship is 0.661. Thus, based on the scheme defined by Schober et al. (2018), the correlation described by the model is considered to be strong."

Line 207 (revised): Added "For the purposes of this paper, we collected the parameters of the best-fit, the root mean squared error (RMSE), and the residual deviation (S) for the run" |
| 1 | Line 220: It doesn't appear this way from the observations in Figure 4, but do the model studies show any evidence that the signal gets saturated (that is, is there a point where PWV is so high that any additional PWV can't be detected from the brightness temperature observations)? | The model studies might be expected to show this saturation for unrealistically high PWV, but we have not explored this parameter space and no measurements have been made in sufficiently high PWV for saturation to be observed. | No changes were made in response to this comment |
| 1 | Line 257. This cost info is very important and should appear in the intro. | We have added this information to the introduction. | Line 65 (revised): Added "(under $50 USD)"

Line 328 (markup): Removed "(under $50 USD)" |

| 2 | A) Please further emphasize the novelty element of this work with respect to the previous related one, both at the end of the Introduction and in the Conclusions. | Reviewer #1 had a similar suggestion, and we have addressed this issue in both introduction and conclusion | Line 65-73 and 294-300 (revised): Added details on the novelty of the paper. |
|---|---|---|---|
| 2 | B) I am concerned about the TPW dataset, given the large distance between the measuring sites. Could you add any other source, closer to the site of interest? | We have investigated other measurement sites (SUOMINET and AERONET, please see response to reviewer #1) that utilize a variety of different measurement techniques and have found that the NWS radiosondes are the most reliable sources of data available. We have added a discussion in the manuscript and a figure in the appendix to further address this issue. | Line 346-352 (revised): Added Appendix B to compare the SuomiNet, AERONET, and radiosonde data sources. |
| 2 | C) Moreover, while I do not know if this is feasible and meaningful here, I reckon that separating your dataset into training/test subsets would be beneficial for this work, so that you could evaluate the fit on an independent dataset via the standard statistical analysis, hence improving the overall quality of this paper. | We have implemented an 80/20 training and testing split into our analysis. Our new figure 4 includes the best-fit line generated by the training set and our analysis looks into the results of our testing set to evaluate the fit. We have updated the appropriate sections to reflect these changes. | Line 204 (revised): Added "For the data partition, we split the data such that 80% was dedicated to training the regression model and the remaining 20% is for evaluating and testing the model"

Section 3.3 (revised): Added additional discussion of iterative method and partition to reflect new analysis method.

Figure 4, 6 (revised): Revised plots that apply new analysis method |
| 2 | D) I wonder whether the –50 degrees instrument threshold has not been too strict a limit in this work and related measurements. Please add a few statements explaining why this has (not) been a limiting issue in your work. | The -50 degrees threshold is a limitation built into the sensor itself, not something we have control over.  The paper has been revised to clarify this point. Fortunately, we have not seen a significant number of AMES measurements exceed the instrument threshold so this has not been a major concern. | Line 82 (revised): Added "hardware-imposed"

Line 88 (revised): Changed "Finally, the AMES thermometer can measure temperatures from" to "The AMES thermometer has a low temperature measurement limit of -50$^{\circ}$C and an |

| | | | upper limit of 550$^{\circ}$C." |
|---|---|---|---|
| 2 | E) Finally, I believe this is a paper on the retrieval of Integrated Water Vapor (IWV), since all measurements are in clear sky. If so, I would suggest rephrasing through the whole manuscript. | Per the recommendation of reviewer#1, we have updated the paper to use the term Precipitable Water Vapor (PWV) rather than Total Precipitable Water (TPW) or IWV. | We have changed TPW to PWV throughout the paper. |
| 2 | Line 5: "We have analyzed this relationship: what relationship are we talking about? Please amend accordingly. | This refers to the relationship between zenith clear sky temperature and PWV. The paper has been revised to clarify this point. | Line 4 (revised): We have updated the paper to say "We have analyzed relationships between PWV and zenith sky temperature measurements for the dry…" |
| 2 | Line 10/11: "but with parameters that are different than those obtained for the Gulf Coast". What are you referring to? Please add detail. | We were referring to the North American Gulf Coast (Texas), the location of the Mims et al measurements. We have updated the document to explicitly state this. | Line 10 (revised): Changed to "but with parameters that are different than those obtained for the previously over the more moist climate zone of the North American Gulf Coast". |
| 2 | Line 36: I suggest replacing "Under clear skies that are the focus of our work" with "In clear sky (the focus of this work)," | Change implemented in revised manuscript. | Line 32 (revised): Replaced "Under clear skies that are the main focus of our work" to "In clear skies (the focus of this work)," |
| 2 | Line 51: Please provide an adequate number of references for each method. | We have added citations to this section of the manuscript. | Line 45-47 (revised): Added Guan et al., 2019; Li et al., 2003; Means and Cayan, 2013; Bevis et al., 1994; Raj et al., 2004; Thome et al., 1992; Thomason, 1985; Liljegren, 1994; Hogg et al., 1983. |
| 2 | I suggest to completely remove the TE1610 sensor discussion from the paper, as I understand it has been no use for this work. | Discussion of the TE1610 sensor has been removed. | Line 96 (markup): Removed "of -20C through 537C. Attempts to determine the infrared wavelength band that this sensor operates in were inconclusive due to the lack of clear sky data available. The error for temperature readings, as determined by |

| | | | the manufacturer, is 2.5C." |
|---|---|---|---|
| 2 | Line 174-175: I find this statement redundant, and overall, I am not expecting to see any type of results from the FLIR instruments, given that you decided not to include any. Again, in line 195, I guess there's no need to mention it. I suggest keeping only the discussion about instruments/dataset effectively used in the end, as this would make the work neater and improve its flow. However, the discussion in the appendix is just fine, as it "proves" the reason why FLIR3 was not used. | Discussion of the FLIR has been condensed to reflect the reviewer's comments. | Line 250 (markup): Removed "and 3.9C for FLIR i3 and AMES respectively"

Line 218 (markup): Removed "Since the FLIR i3 may not produce reliable measurements below its temperature threshold, we have assigned these temperature measurements as not-a-number, and thus are not processed in the final analysis" |

---

## Author Response (AR2)

|   | In the end, this work relies on observations
that are frequently 6 h old (or 6 h early), at
least 100 km away, and much more moist than
more local observations. Using the radiosonde
observations may be the appropriate course of
action, but it needs to be demonstrated that
this set of decisions is the correct one.                                                                                                                                                                                                                                                                                                                                                                                                  |                                                                                                                                                                                                                                                                                                                                                                                                                                                                                                                                                                                                                                                                                                                                                                                                                                                           | Line 183 (markup): removed
"Appendix B compares out derived
PWV with both SuomiNet and
AERONET observations for one year
(2020)"
Line 184 (markup): added "The
corresponding linearized weighting
factors are 0.75 for ABQ and 0.25 for
EPZ"
Line 186 (markup): changed "all three
datasets are" to "PWV data from
SuomiNet, AeroNet, and radiosonde
means are all"
Line 187 - 193 (markup): changes to
discussion on the data gaps for
SuomiNet and AERONET.
Line 252 - 259 (markup): Added
additional discussion on SuomiNet and
AERONET analysis.
Line 376 - 384 (markup): removed
Appendix B.
Figure B1: removed. |
|---|-----------------------------------------------------------------------------------------------------------------------------------------------------------------------------------------------------------------------------------------------------------------------------------------------------------------------------------------------------------------------------------------------------------------------------------------------------------------------------------------------------------------------------------------------------------------------------------------------------------------------------------------------------------------------------------------------------------------------------------------------|-----------------------------------------------------------------------------------------------------------------------------------------------------------------------------------------------------------------------------------------------------------------------------------------------------------------------------------------------------------------------------------------------------------------------------------------------------------------------------------------------------------------------------------------------------------------------------------------------------------------------------------------------------------------------------------------------------------------------------------------------------------------------------------------------------------------------------------------------------------|--------------------------------------------------------------------------------------------------------------------------------------------------------------------------------------------------------------------------------------------------------------------------------------------------------------------------------------------------------------------------------------------------------------------------------------------------------------------------------------------------------------------------------------------------------------------------------------------------------------------------------------------------------------------------------------|
| 1 | Finally, as I read through this work again, I'm
left with one very fundamental question: how
good is it? An analysis that shows the
relationship for the IR PWV product to some
kind of truth (be it the merged sondes,
AERONET, etc.) seems to be lacking. The
figures shown in the present work, such as the
relationship between the sky temperature and
PWV, are important but the relationship
between the new product and the truth is
critical. This could take the form of a
scatterplot, histogram of the differences in various
PWV bins, etc., but something should be in
there. Crucially, I do not have a sense of how
well the product performs as a function of
different values. | For a journal that is focused on
measurement techniques, it does not
seem appropriate to refer to "truth" as
opposed to actual measurements with
their known uncertainties and possible
biases. We believe that the integrated
humidities from radiosondes likely
provide the most accurate PWV
compared to GPS or sun photometer
measurements. However, the spatial
and temporal interpolation to the
Socorro location likely involves a
higher degree of uncertainty for the
weighted sonde means. In the end,
there is simply not enough AERONET
data with which to base our analysis,
and SuomiNet has too large of an
elevation offset. On the other hand,
we appreciate the suggestion of
adding scatter plots for comparison.
We have added a figure (Figure 6) as
discussed above. |                                                                                                                                                                                                                                                                                                                                                                                                                                                                                                                                                                                                                                                                                      |
| 1 | 98. For the observations taken at 2300 UTC, are they matched to temporally averaged radiosonde observations or are they just matched to the nearest sonde time?                                                                                                                                                                                                                                                                                                                                                                                                                                                                                                                                                                               | We consistently use the weighted average of PWV across all of our observation comparisons.                                                                                                                                                                                                                                                                                                                                                                                                                                                                                                                                                                                                                                                                                                                                                                | No changes                                                                                                                                                                                                                                                                                                                                                                                                                                                                                                                                                                                                                                                                           |
| 1 | 112. It is important to emphasize that the determination of clear or cloudy skies is a subjective observation by a human observer.                                                                                                                                                                                                                                                                                                                                                                                                                                                                                                                                                                                                            | We have revised the paper to clarify that the observations are subjective.                                                                                                                                                                                                                                                                                                                                                                                                                                                                                                                                                                                                                                                                                                                                                                                | Line 113 (markup): added subjective                                                                                                                                                                                                                                                                                                                                                                                                                                                                                                                                                                                                                                                  |
| 1 | 115. The lack of brightness temperature
observations below a given temperature
threshold (resulting in NaN values) means that
low PWV values cannot be observed with this
method. What is the minimum PWV value that
can be observed, and what is the seasonal
distribution of missing data? This seems like
an important issue that end users ought to be
aware of. I assume that this is a more frequent
occurrence in the high deserts of New Mexico
than it is in the environment observed by
Mims, and that wintertime values are more
likely to be missing, but these points should be                                                                                                              | As specified in the paper, the primary
AMES sensor has a lower
temperature threshold of -50 degrees
C, and this results in NaN values less
than 4% of the time (Table 1). As
expected, the lowest temperatures
occur during the coldest part of the
winter. The low-temperature
threshold limits minimum PWV to
approximately 3 mm, as seen in Table
1 and in Figure 5.                                                                                                                                                                                                                                                                                                                                                                                                                                                     | No changes                                                                                                                                                                                                                                                                                                                                                                                                                                                                                                                                                                                                                                                                           |

|   | made explicit in the text.                                                                                                                                                                                                                                                                                                                                                                                                                                                                                                                                                                                                                                                                                                                                     |                                                                                                                                                                                                                                                                                                                                            |                                                                                                                                                                                                                                                                                                                                                                                                                                                                                |
|---|----------------------------------------------------------------------------------------------------------------------------------------------------------------------------------------------------------------------------------------------------------------------------------------------------------------------------------------------------------------------------------------------------------------------------------------------------------------------------------------------------------------------------------------------------------------------------------------------------------------------------------------------------------------------------------------------------------------------------------------------------------------|--------------------------------------------------------------------------------------------------------------------------------------------------------------------------------------------------------------------------------------------------------------------------------------------------------------------------------------------|--------------------------------------------------------------------------------------------------------------------------------------------------------------------------------------------------------------------------------------------------------------------------------------------------------------------------------------------------------------------------------------------------------------------------------------------------------------------------------|
| 1 | Figure 1: This is an extremely minor point, and you can address or ignore as you see fit, but I find figures easier to interpret when grid lines are present.                                                                                                                                                                                                                                                                                                                                                                                                                                                                                                                                                                                                  | We have decided to keep the plots
without gridlines to improve readability
on data-heavy figures.                                                                                                                                                                                                                                    | No changes                                                                                                                                                                                                                                                                                                                                                                                                                                                                     |
| 1 | 142. When you say ground temperature, do you specifically mean skin temperature as measured by the IR thermometer?                                                                                                                                                                                                                                                                                                                                                                                                                                                                                                                                                                                                                                             | Ground temperature and skin
temperature as measured by the IR
thermometer are effectively the same,
as now pointed out in the manuscript.                                                                                                                                                                                         | Line 105 (markup): added "(the effective IR skin temperature)"                                                                                                                                                                                                                                                                                                                                                                                                                 |
| 1 | 165. If the Suominet and AERONET
observations are going to be part of this
analysis (even if only in the appendix), their
locations should be noted on Fig. 2.                                                                                                                                                                                                                                                                                                                                                                                                                                                                                                                                                                                        | We have added the SuomiNet and AERONET locations to the map.                                                                                                                                                                                                                                                                               | Figure 2 (revised): Changed to include
Suominet and AERONET locations
and is now colorblind friendly. The
caption was also adjusted.
Line 171 (markup): added "Socorro,
ABQ, EPZ."
Line 171 (markup): added ", along with
the locations of Socorro, SuomiNet,
and Sevilleta AERONET sites."                                                                                                                                                            |
| 1 | 165. Sometimes the text refers to Figure N,
and other times it refers to Fig. N. This may be
a stylistic choice, as it appears that the word is
spelled out at the start of a sentence but not
elsewhere, so I don't know how much
consistency you are going for here.                                                                                                                                                                                                                                                                                                                                                                                                                                                                          | This style choice is based on the formatting guidelines laid out by the journal.                                                                                                                                                                                                                                                           | No changes                                                                                                                                                                                                                                                                                                                                                                                                                                                                     |
| 1 | 174-175: I'm not seeing where your product
appears in Appendix B (unless you only mean
the merged sondes). This goes back to the
point I made in the major comments above
about not really getting a sense of the skill or
utility of the product.                                                                                                                                                                                                                                                                                                                                                                                                                                                                                              | Given that we have plotted 5 datasets
in the time series plot (which now
appears in the manuscript rather than
the appendix), we have not added our
PWV IR product to this plot. Instead,
we now compare the PWV IR product
to SuomiNet and AERONET in Fig. 6.                                                           | No changes                                                                                                                                                                                                                                                                                                                                                                                                                                                                     |
| 1 | 203. This seems like a counterintuitive way to
approach the exceedance thresholding, as
though the most important thing was to
preserve 90% of the dataset instead of crafting
a representative dataset. If the data are
unrepresentative, they should not be used
regardless of how many event dates must be
removed. At a minimum, it is important to
know how many standard deviations that 55%
difference represents. (Also note: in the
response to the reviewers, the authors stated
this was a 75% threshold, so they should
verify which value is the correct one.) It is
easier to scientifically justify a
standard-deviation-based filter than a filter
designed to preserve a certain fraction of the | The 75% value was a typo and the correct value was 55%. We apologize for the confusion. To address this comment we have redesigned this feature to compare the standard deviation of the PWV measurements for the individual days with the average standard deviation over all days. The paper has been adjusted to reflect these changes. | Line 232 (markup): changed "A = 20.2
mm and B = 0.036" to "A = 18:48 mm
and B = 0:034"
Line 217 (markup): changed "relative
difference" to "standard deviation"
Line 218 (markup): changed "individual
PWV observations to the daily mean of
both ABQ and EPZ" to "the standard
deviation of the PWV observations for
a given day with the mean of the daily
standard deviations over the entire
dataset."
Line 220 (markup): changed "any |

|   | total dataset, even if in the end you tune one filter to match the other.                                                                                                                                                                                                                                                                                                                                                                                                                                                                     |                                                                                                                                                                                                                                         | difference exceeds a fixed 55%. This
threshold value was determined so
that no more than 10% of the days are
rejected by this filter, while still
ensuring" to "the standard deviation is
more than twice the overall mean
value" |
|---|-----------------------------------------------------------------------------------------------------------------------------------------------------------------------------------------------------------------------------------------------------------------------------------------------------------------------------------------------------------------------------------------------------------------------------------------------------------------------------------------------------------------------------------------------|-----------------------------------------------------------------------------------------------------------------------------------------------------------------------------------------------------------------------------------------|-----------------------------------------------------------------------------------------------------------------------------------------------------------------------------------------------------------------------------------------------------|
|   |                                                                                                                                                                                                                                                                                                                                                                                                                                                                                                                                               |                                                                                                                                                                                                                                         | Line 223 (markup): changed
"radiosondes do not bias" to "PWV, or
between 00Z and 12Z observations,
do not negatively impact"                                                                                                               |
|   |                                                                                                                                                                                                                                                                                                                                                                                                                                                                                                                                               |                                                                                                                                                                                                                                         | Line 224 (markup): Added
"Approximently 12% of the days are
rejected by this filter"                                                                                                                                                          |
|   |                                                                                                                                                                                                                                                                                                                                                                                                                                                                                                                                               |                                                                                                                                                                                                                                         | Line 247 (markup): changed "mean" to
"standard deviation"                                                                                                                                                                                        |
|   |                                                                                                                                                                                                                                                                                                                                                                                                                                                                                                                                               |                                                                                                                                                                                                                                         | Line 250 (markup): changed "3.79" to "3.64"                                                                                                                                                                                                         |
|   |                                                                                                                                                                                                                                                                                                                                                                                                                                                                                                                                               |                                                                                                                                                                                                                                         | Line 260 (markup): changed "3.75" to
"3.60"                                                                                                                                                                                                      |
|   |                                                                                                                                                                                                                                                                                                                                                                                                                                                                                                                                               |                                                                                                                                                                                                                                         | Line 262 (markup): changed "4.52" to
"4.63"                                                                                                                                                                                                      |
| 1 | 226. This is close to what I was suggesting
when I suggested a monte carlo simulation.
My thinking was that you could take an
IR-observed temperature, randomly perturb it
by some value drawn from a gaussian, and
plug that into your tool to obtain a new PWV.
Do that a few thousand times, and you'll have
an estimate on how the instrument
uncertainties contribute to uncertainties in
PWV. This doesn't take into account the
uncertainties in the model, however, which
your approach seems to do. | We appreciate your feedback.                                                                                                                                                                                                            | No changes                                                                                                                                                                                                                                          |
| 1 | 230. Does this RMSE vary with the magnitude
of the signal? Looking at Figure B1, a RMSE
of 0.35 cm is very close to the observed value
for the winter months. Do you expect that the
error bars are very similar throughout the year,
or do the larger PWV values in the summer
have greater uncertainties associated with
them?                                                                                                                                                                                         | Although it is possible that the RMSE
contains a seasonal component, we
feel that a detailed analysis of
seasonal changes in RMSE in the
paper is not not warranted at this
point, given two years of available
data. | No changes                                                                                                                                                                                                                                          |
| 1 | 285. This seems to imply that it may be
possible to derive the appropriate relationships
between PWV and the IR temp without                                                                                                                                                                                                                                                                                                                                                                                                            | Based on the theory alone, it is not
possible at this time to derive accurate
fit coefficients for all locations, and it                                                                                                          | No changes                                                                                                                                                                                                                                          |

|   | needing to take two years of manual
observations to generate a testing dataset. Is
this true? Or, rather, are there ways to arrive at
the needed coefficients using existing data?
(Earlier, when I said that I wanted to point my
IR thermometer at the sky to get PWV, I meant
it.) In all seriousness, you have done a good
job demonstrating that the system needs to be
trained to specific locations due to the large
climatological variability in water vapor
content. But are there ways to achieve
acceptable results using a priori data? I think
this is an important point for the issues raised
in the conclusions, as substantial datasets will
need to be collected by citizen scientists and
school groups just to train the relationships. If
an initial model can be implemented
immediately from prior observations, NWP,
etc., the adoption of such a program will likely
increase. | will likely be necessary to build up a
minimum database in order to carry
out a widespread citizen science
program. One step in this process is
to explore the parameter space for a
diverse set of locations and
meteorological conditions. We feel
that the current paper is an important
step towards that goal. |                                   |
|---|-------------------------------------------------------------------------------------------------------------------------------------------------------------------------------------------------------------------------------------------------------------------------------------------------------------------------------------------------------------------------------------------------------------------------------------------------------------------------------------------------------------------------------------------------------------------------------------------------------------------------------------------------------------------------------------------------------------------------------------------------------------------------------------------------------------------------------------------------------------------------------------------------------------------------------------------------------------------|---------------------------------------------------------------------------------------------------------------------------------------------------------------------------------------------------------------------------------------------------------------------------------------------------------------------------------------------|-----------------------------------|
| 1 | Figure B1. I keep coming back to this figure
throughout reading and reviewing this paper.
At times I wonder if this figure is important
enough that it deserves promotion ot the main
body of the paper.                                                                                                                                                                                                                                                                                                                                                                                                                                                                                                                                                                                                                                                                                                                                              | We appreciate this suggestion, and
have integrated Appendix B into the
main paper along with additional
discussion as noted above.                                                                                                                                                                                                 | Changes discussed in the top row. |